# Text Input in Virtual Reality Using a Tracked Drawing Tablet

FirstName Surname†
Department Name
Institution/University Name
City State Country
email@email.com

FirstName Surname
Department Name
Institution/University Name
City State Country
email@email.com

FirstName Surname
Department Name
Institution/University Name
City State Country
email@email.com

## ABSTRACT

We present an experiment evaluating the effectiveness of a tracked drawing tablet for use in virtual reality (VR) text input. Participants first completed a text input pre-test, entering several phrases using a regular keyboard in the real environment. Participants then entered text in VR using an HTC Vive system, with a Vive tracker mounted onto a drawing tablet using a QWERTY soft keyboard overlaid on the virtual tablet. This was similar to text input using stylus-supported mobile devices in non-VR contexts. We experimentally compared this text entry method to using a Vive controller with ray-casting to point at a virtual QWERTY keyboard. Our results indicate that not only did participants prefer the Vive controller, it also offered superior entry speed (16.31 wpm vs. 12.79 wpm with the tablet and stylus). Notably, our analysis failed to detect a significant difference in error rate between the two VR text input methods. Pre-test scores were also correlated to measured entry speeds, and reveal that user typing speed on physical keyboards provides a modest predictor of VR text input speed ($R^2$ of 0.6 for the Vive controller, 0.45 for the tablet).

## KEYWORDS

Text input, virtual reality.

## 1 Introduction

Virtual reality (VR) devices are becoming increasingly affordable and performant. Between the falling prices, and the recent emergence of wireless head-mounted displays (HMDs), VR is also becoming more accessible. Despite these recent advancements in interaction technologies for VR systems, an ongoing problem is symbolic and text input in VR. Text input has traditionally received somewhat less attention from the research community than other interaction tasks in VR [3]. This is likely because substantial text input has been a more "niche" task than selection, manipulation, or navigation. Though the exact reasons for this are unclear, it may be related to lower quality VR systems of the past that were uncomfortable to use

for lengthy composition. In the past few years, however, there have been several studies on different text entry methods in VR [11, 12, 34, 41], suggesting increasing application demand.

Though replacing the standard, physical keyboards for heavy text-entry tasks (e.g. writing a paper or writing programming code) would be difficult. The general lack of physical embodiment and difficulty using keyboards in midair have given rise to alternative techniques to address the many use cases of short, yet arbitrary, text input. Consider, for example sending quick SMS-like messages to another user or annotating part of the environment during a design review in VR, or calling up a webpage by typing a URL. It is unattractive to have to switch context from VR (particularly when using HMDs) to a physical keyboard in order to perform such tasks. We thus explore approaches that can be used for VR text entry that are easier to use for a novice user, and which offer acceptable performance levels. In this study, we compared the most common method of text input in VR – using a 3D tracked controller employing ray-casting – to text input using a tracked tablet with a stylus.

Using a pen and stylus with an on-screen QWERTY keyboard is common in the mobile computing domain. Many modern smartphones and tablets include a stylus (e.g., Samsung's Galaxy Note line), which among other operations (e.g., drawing) can be used with onscreen keyboards to support text input. Using the stylus has the potential for better performance than fingers due to the "fat finger" problem [37]. This style of interaction is naturally familiar from writing with a pen and paper, and it has been used in VR before [4, 10, 32]. We propose to leverage this familiarity with modern VR hardware by adding a tracker to a digital drawing tablet while using the tablet's digitizer to detect the stylus contact point.

Previous work has taken a similar approach, using a 3D tracked physical pen and tablet metaphor [4], or using a wooden tablet and pen [10]. Our approach is based on the observation that simply tracking a drawing tablet yields higher precision on the contact point than using a secondary tracker on the stylus while also keeping the stylus unencumbered. Moreover, the tablet (and Vive tracker) are relatively inexpensive compared to light-weight optical trackers (e.g., Vicon) that could be used to track a stylus. The result is similar to text input on mobile devices; the main difference between our scenario and mobile devices is that the user is sitting in a

*Article Title Footnote needs to be captured as Title Note

†Author Footnote to be captured as Author Note

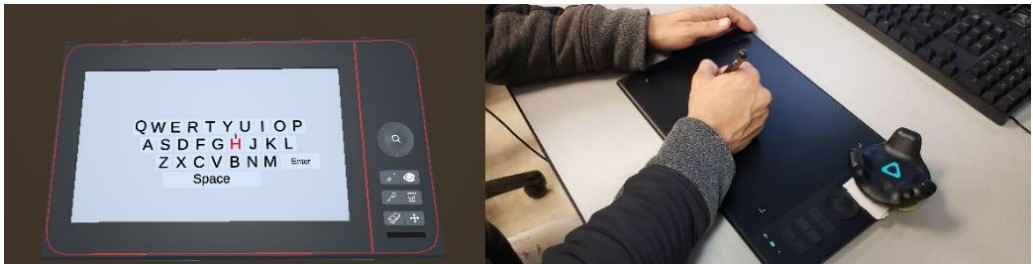

**Figure 1: (Left) The VR view, showing the QWERTY soft keyboard. The "H" key is highlighted to reflect that the stylus is currently hovering over it. (Right) Participant performing the experiment with the Tablet and Stylus.**

virtual environment instead of the real one. In our experiment, participants used the tracked tablet in VR and entered text using a stylus by selecting characters on a virtual QWERTY keyboard displayed on the tablet screen in VR. See Figure 1.

The rest of the paper is structured as follows: Section 2 describes previous work on tablets in VR and the state of the art in VR text entry. Section 3 describes our experimental methodology. Sections 4 and 5 present our results and summarize design guidelines for future VR text entry methods.

## 2   Literature Review

Using a tablet and stylus for text entry has been studied extensively in the HCI community in non-VR scenarios [6, 9, 51, 54]. Entry rates using a stylus for input with a QWERTY soft keyboard range between 8.9 wpm and 30.1 wpm, according to Soukoreff et al. [49]. Here we focus on previous research using tablet devices and different text entry techniques for VR.

### 2.1  Tablets and Mobile Devices in VR

One of the earliest examples of using a tablet-like device in VR was the HARP system developed by Lindeman et al. [22]. Their results showed that a 2D stylus and tablet metaphor used in VR provided better support for precise selection actions. They argued this was a direct result of providing a tactile surface via the virtual tablet. The benefits of so-called "passive" haptic feedback are well-known in the VR research community [1, 5, 8, 15, 27]. They also showed that using a hand-held device is preferable to fixed-position devices, as they provide freedom for working effectively in IVEs.

Poupyrev et al. developed one of the earliest text entry methods in VR, the Virtual Notepad. It was a spatially tracked tablet and stylus system for taking notes in VR [32]. Using the Virtual Notepad users could take notes, modify them, add or remove pages, and manipulate the documents within the VE. The system employed character recognition to detect and issue handwritten commands in the form of individual letters. Medeiros et al. proposed using mobile devices for interaction in virtual environments [36]. They concluded that user familiarity with these mobile devices reduces their resistance to immersive virtual environments (IVEs). Other researchers

have explored the use of mobiles as input devices in VR, for example, to select and manipulate objects [17].

Several other studies have explored the use of mobile devices and tablets in VR contexts [7, 14, 42, 47]. Perhaps the most similar work to ours, Kim and Kim's proposed method, using a smartphone and its hovering function for text entry in VR [18]. HoVR used a smartphone's hovering function for text entry in VR. However, they did not report text entry speed or error rate and reported only task completion time, and also did not use known subsets for the stimulus text [24], which makes comparing results difficult.

### 2.2  Text Entry Methods in VR

Compared to interaction tasks like selection and navigation, there are comparatively few studies on text input in VR. Table 1 presents a summary of several key studies.

Bowman et al. empirically compared four different text entry techniques including a pen and tablet, voice recognition, a one-hand chord keyboard and a method using pinch gloves [4]. Speech recognition was fastest, at around 14 words per minute (wpm). Their pen and tablet metaphor offered entry rates of up 12 wpm. Unlike our study, which employs a tablet digitizer to "track" the stylus, their study used a 3D tracked stylus to touch letters, indicating selection by pressing a stylus button. Another example of using a pen-based approach, González et al. tracked a wooden tablet and stylus with a sensor, although the type of sensor is not reported. In a series of text entry experiments, they report entry rates between 7 and 8 wpm [10] with the stylus and tablet.

Speech recognition is a potentially attractive method of VR text input. The SWIFTER speech recognition system improved on Bowman's speech-based approach [4], achieving average entry speeds of up to 23 WPM [31]. However, although speech to text-based techniques are fast, they are not well-suited to loud environments, or in situations requiring discretion (e.g., users having private conversations, or entering password securely). Also, editing is challenging with speech-based methods; for example, cursor positioning is problematic. Moreover, speech can interfere with the cognitive process required to enter text [38, 43].

| 1st Author | Text Entry Method | Entry Rate (wpm) | Error Rate | Notes |
|---|---|---|---|---|
| Poupyrev [32] | Tablet & stylus with digital ink | Not reported | Not reported | Switched between showing hands and the stylus and used character recognition for text input. |
| Bowman [4] | QWERTY keyboard with pen and tablet metaphor | 10* | 7.14 errors per subject | Original result were reported in character per minute (cpm). *Note: we converted to wpm by dividing the original results by five. |
| | Pinch keyboard | 5* | 43.17 error per subject | |
| González [10] | Pen based QWERTY keyboard | 7* | 7% character error rate | Used tablet and pen made out of wood, tracked via sensor. Users could see the pen. * Note: Entry rate originally reported in CPM, converted to wpm. |
| | Pen based disk keyboard | 4* | 2% character error rate | |
| Grubert [11, 12] | QWERTY desktop keyboard | 26 | 2.1% character error rate | Different hand representations were used. i.e. full hand vs. fingers only. Also looked into repositioning the keyboard in VR. |
| | QWERTY touchscreen keyboard | 11 | 2.7% character error rate | |
| Yu [55] | Head pointed & Gesture based | 10 to 19 | 1.23% to 3.08% corrected error rate | Investigated TapType, DwellType, and GestureType techniques. |
| Yu [56] | Dual joystick controller | 7 to 15 | 1.57% to 1.59 uncorrected error rate | Used a circular keyboard layout. |
| Kim [18] | QWERTY touchscreen keyboard | Not reported | Not reported | Used smartphone's hovering function for finger tracking. Reported task completion time. |
| Kuester [21] | Wearable glove | Not reported | Not reported | Used the concept of column and rows found in traditional keyboards. |
| Rajanna [34] | Gaze typing | 6 to 9 | .02% to .08% rate of back space | Sitting and biking were conditions in the experiments. |
| Xu [53] | Head motions | 8 to 12 | 2.25% to 2.46% uncorrected error rate | Dwell and hands free interaction method, used a circular keyboard layout. |
| Prätorius [33] | Thumb to finger taps | Not reported | Not reported for text entry. | Reported keystroke per character. |
| Gugenheimer [13] | Split QWERTY touchscreen keyboard | 10 | Not reported | Used displayed fixed UIs, users wore a touch sensor on the HMD. |
| Speicher [41] | QWERTY keyboard with Controller Pointing | 15 | 0.97% Corrected error rate | Also looked into the physical demand required and cyber sickness in different text entry methods in VR. |
| | QWERTY keyboard with Controller Tapping | 12 | 1.94% Corrected error rate | |

**Table 1: Several text entry studies in immersive virtual reality environments and their performance results.**

Several studies have explored the use of game controllers as text input mechanisms both in VR, and in other similar use scenarios, such as games [20, 29, 50, 56]. Isokoski et al. used a controller and a tablet with a stylus in their experiments [16]. They reported entry rates ranging from 6 to 8 wpm using the Quickwriting method [16, 30]. Entry speed with conventional game controllers tends to range between 6 to 15 wpm [50, 56].

Other researchers explored the use of head motion and gaze direction for typing in VR [34, 53, 55]. Yu et al. explored three head-based techniques for text entry in VR [55]. They reported average entry speed of 24.73 WPM with their GestureType technique after one hour of training. Gugenheimer et al. introduced FaceTouch, which employed display-fixed UIs [13]. Their approach employed touchpads mounted on the front face of the HMD, which users touched to enter text. In an informal study on text entry using a split QWERTY keyboard, the authors reported average text entry speed of approximately 10 wpm. Rajanna et al. focused on investigating how keyboard design,

selection method, and motion in the field of view impact typing performance and user experience [34]. They concluded that VR gaze typing is viable, if somewhat unnatural.

Several recent VR text input studies have investigated the use of physical keyboards with various hand visualizations [2, 11, 12, 44, 45]. Knierim et al. evaluated the effects of virtual hand representation and hand transparency on typing performance of experienced and inexperienced typists in VR [19]. Their results suggest that experienced users (e.g., touch typists) performance is not significantly affected by missing hands or different hand visualization. However, inexperienced users are impacted by these factors. Similar work by Grubert et al. also investigated methods for virtual hand representation with minimalistic fingertip visualization [11, 12]. Specifically, their minimalistic visualization showed only dots at the fingertips rather than an entire hand visualization. They report that even with minimalistic fingertip representations, entry

speeds ranging from 34 to 38 wpm are possible, depending on hand and finger representation.

Other VR text input methods employing 3D tracked controllers employing direct touch, or ray-casting, to select keys from a virtual keyboard. Entry rate ranged from 12 to 15 wpm [41]. Several other studies employed gloves and hand gestures both in VR and non-VR context [21, 26, 28, 33, 35, 46, 48]. Yi et al. reported entry rates of up to 29 wpm using their hand based method in a non-VR setting [52]. As suggested by Grubert et al., methods that use a controller for text entry tend to have a higher learning curves and require more training than keyboards, and can also cause user fatigue [11].

## 3 Methodology

### 3.1 Participants

We recruited 28 participants from our local community but ended up removing four of them. Two were extreme outliers (entry speed scores more than 3 $SD$s from the mean), and there were logging errors with the other two. This left us with 24 participants upon which our analysis is based. There were 9 female and 15 male participants, aged between 18 and 54 ($M$ = 26.21, $SD$ = 8.19). Eighteen participants reported that they had not played 3D games, or only played them infrequently. Twenty-one participants had very little or no prior experience with VR. Seven participants reported regularly using a pen or stylus for typing on their tablet or smartphone. One participant indicated that they did not text at all, while nine texted frequently during a typical day. The rest of them reported moderate texting. All participants had normal or corrected-to-normal stereo vision, assessed by having participants correctly determine the depth of two spheres presented in the scene.

### 3.2 Apparatus

#### 3.2.1 Hardware

We used a PC with an Intel *Core i7* CPU with an NVIDIA *Geforce GTX 1080* graphics card for both the experiment. We used the HTC *Vive VR* platform, which includes an HMD with 1080 × 1200 per eye resolution, 90 Hz refresh rate, and a 110° field of view. The tablet was an XP-PEN *STAR 06* wireless drawing tablet. Its dimensions were 354mm × 220mm × 9.9mm with a 254mm × 152.4mm active area, and a 5080 LPI resolution. The tablet included a stylus with a barrel button and a tip switch to support activation upon pressing it against the tablet surface. The 2D location of the stylus is tracked along its surface by the built-in electromagnetic digitizer. We affixed a Vive tracker to the top-right corner of the tablet using velcro tape. See Figure 2.

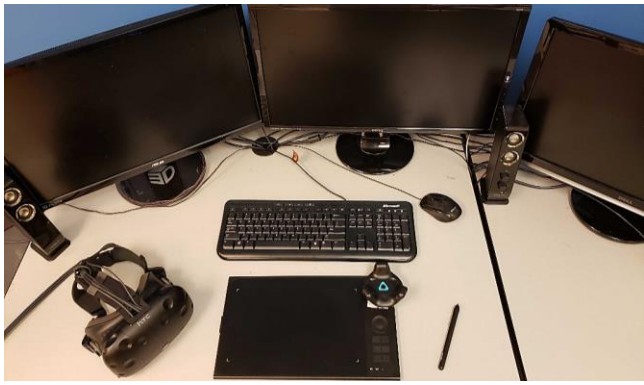

**Figure 2: Tablet and stylus with Vive tracker and Vive HMD along with the physical keyboard used in pre-tests.**

#### 3.2.1 Software

We used MacKenzie's "Typing Text Experiment"[1] software for the pre-test. The pre-test consisted of 30 randomly determined phrases the participants entered using a real keyboard.

The main experiment used our VR software developed in Unity3D. We developed a custom library to get stylus input from the XP-PEN STAR tablet into Unity. The tablet is ordinarily seen as fitting the human interaction device (HID) profile by Windows 10, which, by default would map it to the mouse cursor, which was not desired in VR. To avoid this, we installed a custom LibUSB driver that allowed direct access to the raw data from the tablet. The library provided data such as the coordinate position of the stylus on the tablet surface, the amount of pressure applied by on the stylus tip switch, and whether the stylus was touching the tablet surface or hovering above it within approximately 2 cm.

The software polled the Vive tracker to map a virtual model of the tablet to the physical tablet, co-locating the two. The tablet stylus was used to interact with the tablet. The stylus itself was not tracked, hence tracking was limited to the tip of the stylus in a close range to the tablet surface (about 2 cm). Due to this limitation, we did not render a model of the stylus or hands. However, when the stylus was in the range of the tablet, we displayed a cursor at the stylus tip. Notably, this is how such graphics tablets are typically used, as the display is not collocated, which has the advantage of not covering part of the drawing with the hand. By applying pressure on the stylus tip switch by pressing it against the tablet surface, input events were detected for corresponding keys on the tablet and the corresponding character was entered.

Participants sat in the virtual room seen in Figure 3. Participants were always presented with a simulated QWERTY soft keyboard displayed centred on the tablet (see Figure 1). The tablet was positioned on a table as seen in Figure 3. The current target phrase appeared near the virtual keyboard to reduce the need for glancing during entry. As participants

---

[1] Available at http://www.yorku.ca/mack/HCIbook/

entered the phrase, each keystroke was presented, giving them immediate feedback. While hovering on keys, the letters changed colour to indicate which would be selected if the tip switch was pressed. Upon pressing a key in this fashion, an auditory "click" sound was played and the key letters change colour to yellow. The SPACE bar and the ENTER key each had distinct button press sounds.

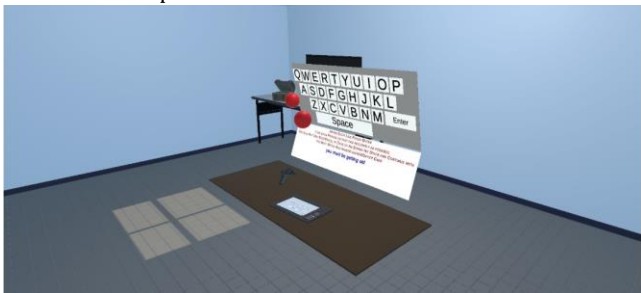

**Figure 3: View of the virtual room seen by participants. Red were spheres used for stereo viewing test.**

Figure 4 depicts the other text entry method, the Vive controller. With this text entry method, participants pointed a ray from the Vive controller at the desired key and pressed the trigger button to select. Upon being intersected by the selection ray, a key changed colour from blue to yellow. Upon pressing the trigger, the selected key would turn into grey. The sound effects were the same as the tablet condition.

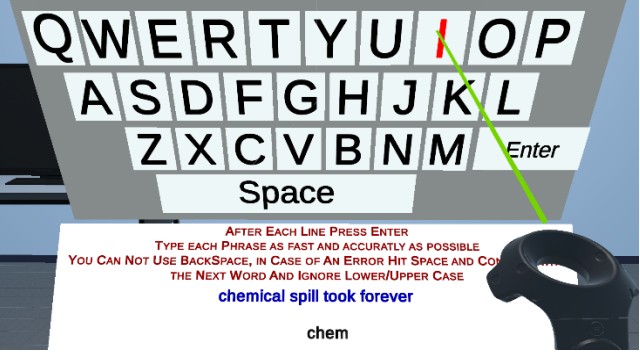

**Figure 4: Vive Controller with ray-casting and soft keyboard.**

## 3.3 Procedure

Before they began, participants first read and signed consent forms and completed a pre-questionnaire to gather demographic data such as age and gaming/VR experience, stylus usage, and mobile text input habits. The experimenter then explained the procedure. Correction (i.e., backspace) was disabled with both the real keyboard during the pre-test as well as the virtual keyboard during the experiment. Auto-correction or similar features were not implemented. Participants were instructed to enter each phrase as quickly and accurately as possible, ignoring any mistakes and pressing the ENTER key to

end each trial. Timing started as soon as the participant entered the first character and stopped as soon as they hit the ENTER key.

Upon starting the experiment, participants first entered 30 phrases using MacKenzie's "Typing Text Experiment" software with real keyboard as a pre-test. After the pre-test, the experimenter demonstrated how to use the Vive controller and the tablet. The participants then entered VR, and were screened for stereo viewing prior to continuing the experiment. They were presented with two red spheres at different depths (Figure 3). They were instructed to reach out and intersect the Vive controller with the spheres to make them disappear. All participants were able to reliably detect the depth of the spheres and hence passed this screening.

The experiment task then started without any training with the interaction devices. Participants were instructed to enter a random pool of phrases from a phrase set commonly used in text entry experiments [24]. Participants were presented with one line from the phrase set at a time.

Participants entered 30 phrases with each text input method (both in pre-test and in VR). After completing 30 phrases for each method in VR, participants completed a questionnaire related to their experience of that method. In the end, they completed a post-questionnaire about their preferred text input method and any comments or suggestions they had.

## 3.4 Design

Our experiment employed a within-subjects design with two independent variables:

> ***Text Input Method***:  Tablet and stylus, Vive Controller
> ***Trial***:  1, 2, 3, …, 30

Text input method order was counterbalanced by having half the participants use the Vive Controller then Tablet and stylus, and the other half in the reverse order. Entry speed, in words per minute (wpm), was calculated as:

$$wpm = \frac{|s|}{T} \times 60 \times \frac{1}{5}$$

where $T$ is the text entry time in seconds, and $|s|$ is the input string length (in characters). This used every five characters (including spaces) as a single word, consistent with the text input literature [25].

Error rates were calculated using character error rate (CER), calculated as:

$$CER\% = \frac{MSD\ (stimulus\ text, response\ text)}{CharLenght(stimulus\ text)} \times 100$$

CER is the minimum number of character-level insertion, deletion, and substitution operations required to transform the response text into the stimulus text, i.e. Minimum String Distance (MSD) between the two, divided by the number of characters in the stimulus text [23, 39]. This metric has the advantage of more correctly representing errors. For example, insertion of a single character early in a phrase results in a single error, rather than a "cascade" of mismatched characters [40]. CER is expressed as a percentage of errors in the original presented text to participants.

## 4  Results

We used repeated-measures ANOVA in all cases. The assumption of sphericity was met in all cases based on Mauchly's test. We were also interested in determining if the participant's touch-typing speed was related to their VR text entry speed. To this end, we employed linear regression to see if they were correlated.

### 4.1  Performance

Figure 5 depicts the average entry speed for each text entry method. The tablet and stylus had a mean entry speed of 12.79 WPM ($SE$ = .71). In contrast, the Vive controller offered faster entry speed at 16.31 WPM ($SE$ = .44). We found a significant main effect of text entry method for entry speed ($F_{1,23}$ = 22.34, $p < .001$, $\eta_p^2 = .49$).

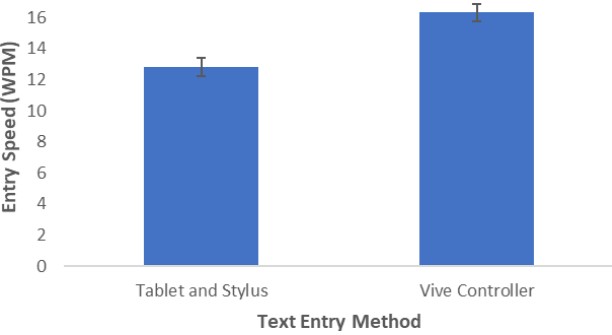

**Figure 5: Average text entry speed for each VR input method. Error bars show 95% confidence interval.**

Figure 6 depicts the average entry speed across each of the 30 trials.Figure 5 The analysis revealed a significant main effect of trial for entry speed ($F_{1,29}$ = 25, $p < .001$, $\eta_p^2 = .52$). The interaction effect between trial and text entry method was not significant for text entry speed ($F_{29,667}$ = .93, ns, $\eta_p^2 = .03$).

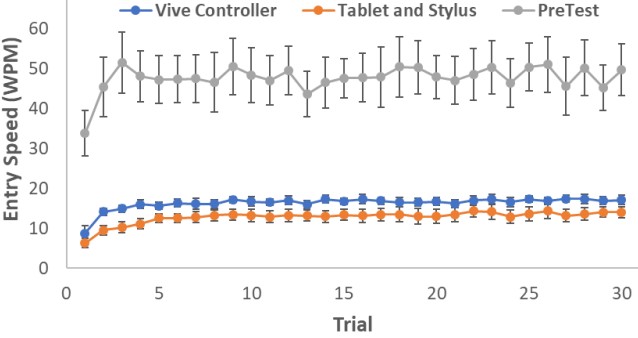

**Figure 6: Average text entry speed in each trial.**

Figure 7 depicts the average error rate in CER for each text entry method. The tablet and stylus technique had a mean error rate of $M$ = 6.42% ($SE$ = 1.66), and the Vive controller had a mean error rate of $M$ = 4.14% ($SE$ =.95). The analysis revealed

that the main effect of text entry method on error rate was not significant ($F_{1,23}$ = 2.04, $p > .05$, $\eta_p^2 = .08$).

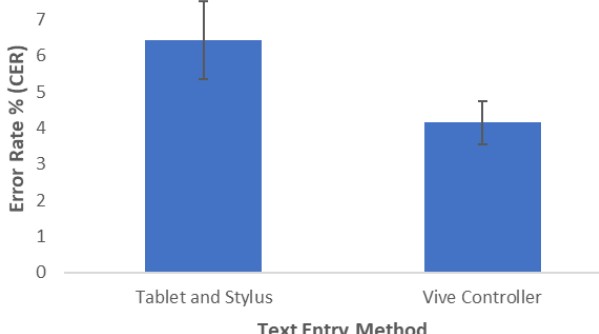

**Figure 7: Average error rate for each VR input method. Error bars show 95% confidence interval.**

Figure 8 shows the error rate for each trial. There was a significant main effect of trial for error rate ($F_{1,29}$ = 1.87, $p = .004$, $\eta_p^2 = .07$). Our analysis also revealed a significant main interaction effect between text entry speed and trial in case of error rate ($F_{29,667}$ = 1.50, $p = .04$, $\eta_p^2 = .06$).

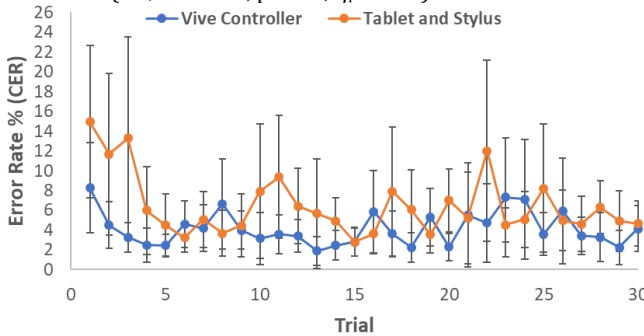

**Figure 8: Average error rate in each trial.**

As seen in Figure 9, there is a modest relationship based on regression analysis. Indeed faster typists on a typical desktop setup had better entry speed with both VR text input methods.

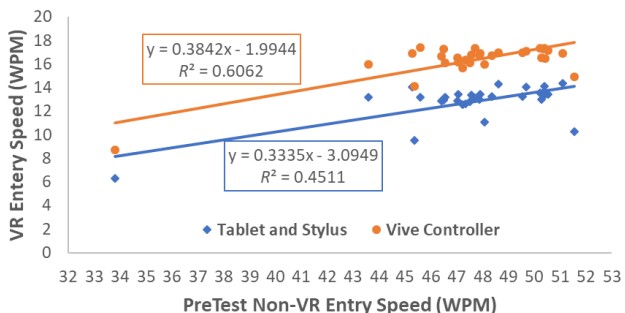

**Figure 9: Linear regression showing the correlation between pre-text entry speeds and VR entry speeds.**

## 4.2 User Experience

All the participants except one preferred the Vive controller text entry method. Few of them had trouble reading the 3D text on the tablet as it was reported that "the text gets quite blurry." Another major disadvantage was the weight of the HMD as it bothered them while using the tablet and that it was hard to look down while wearing the HMD in VR. Participants also reported confusion with the sensitivity of the pen and tablet, they were not sure about the amount of tip switch pressure required to register the input. Most of them found it hard without knowing where the cursor and the stylus were when tracking was lost and reported they wanted to see the stylus at all times. One participant stated that the tablet was "good to use when you get the hang of it, but the distance and specific angle you have to hold the stylus at is a little difficult to get used to". One of them also suggested that implementing two styluses could have improved the speed and efficiency of typing as one could at least be used for spacing.

## 5    Discussion

Our performance results indicated that our proposed method underperformed relative to the Vive controller. Nevertheless, we believe there is some potential for the tablet and stylus to be considered a viable text entry method in VR. While the Vive controller offered faster text entry speed (16.32 wpm vs. 12.80 wpm), we speculate that adding the stylus and hand visualization and improving the tip switch sensitivity could improve the performance of the tablet method further. The tablet and stylus performance is also comparable to several previous text input methods, as seen in Table 1. In particular, its performance is comparable to Bowman's tracked stylus and tablet [4], several game controller methods [56], head motion [53] and touchscreen typing on a soft keyboard [12]. It performs *better* than several previous techniques, including an alternative tracked tablet and stylus [10] and gaze typing [34].

Some participants leaned forward and moved very close to the tablet with their head down to tap on the tablet; this could have caused the HMD to move slightly on their head and cause the blurry effect they reported while using the tablet. Some participants also reported noticing the weight of the HMD only while using the tablet. This also might have to do with the fact that they were hunching over the tablet. In contrast, with the Vive controller condition, they held their heads up to see the soft keyboard. Participants also reported their arms getting tired because of tapping. This was mainly due to more arm movements required to navigate the tablet keyboard for character keys. While using the Vive controller with ray-casting, participants could easily navigate the keyboard by small wrist movements. We believe this could be improved by implementing a swiping text entry method for the tablet, or by using a different soft keyboard layout. Notably, the Vive controller can also be used bimanually, which could potentially increase typing performance. However, participants liked the tactile feedback provided by the tablet, in line with previous work on haptics in VR [5, 15, 27].

## 6    Conclusion

In this paper, we evaluated the effectiveness of using a tablet with a stylus for text input in VR. We designed and developed a text entry method using a QWERTY virtual keyboard on a tablet in VR. Our experiment compared text entry using a tablet and stylus with a QWERTY soft keyboard to using a Vive controller with ray casting. Our study showed that the Vive controller performed better, and was preferred by participants. We also found that VR text input speed can be – to a modest extent – predicted by touch typing entry speed with a conventional keyboard. We argue that there is some potential to improve the tablet, which was roughly as accurate as the Vive controller. Also, if users are already using a tablet for other functions in VR, like a VR design session, using the tablet instead of a controller for text entry is preferable and feels more natural.

## 7    Limitations and Future Work

For our future work, we will improve our design by adding stylus and hand tracking. Using a different keyboard layout design, i.e., a circular keyboard, and swipe typing is also of interest. We are also considering using a new HMDs with higher resolutions, lighter designs, and a wider field of view.

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
