# OpenReview forum: "Text Input in Virtual Reality Using a Tracked Drawing Tablet "
_graphicsinterface.org/Graphics_Interface/2020/Conference — Submitted to GI 2020_

### Official Review · AnonReviewer3 · 2020-02-12
**Literature review was interesting to read; but there were some issues in the study design**

**Rating:** 4
**Confidence:** 3

**Review:**

This work presented a study which investigated the text input performance in VR. Two input method was compared: 1) using a tracked tablet & stylus, and 2) using a VR controller with ray-casting enabled.

In general, the writing was smooth and I found the literature review (incl. Table 1) valuable and would certainly help researchers in this field. I also found some qualitative results very illustrating, e.g., some participants mentioned that extra fatigue was caused by moving their head up and down during the tablet & stylus session.

Citing several existing work which used similar approach, this work proposed that some prior work failed to report text entry speed or error rate (Sec. 2.1). I doubt whether the motivation is convincing (unless the interaction techniques involved is novel or the study is designed and conducted thoroughly).

In addition, several important details about the study conditions were missing: dimensions of the virtual keyboard on the tablet and the ray-casting keyboard.

Only dimensions of the tablet were reported, but given Fig. 1, the keyboard size is still hard to get. The size & depth of the ray-casting keyboard (or FOV of each key) also affects the difficulty of typing in VR. Without these details, it would be hard to compare the results with other work and thus the entry speed or error rate would provide limited value.

In summary, the literature review and the study analysis was good, but given the issues found in the study procedure, I found the contribution was rather incremental. I would argue for a weak reject.

---

### Official Review · AnonReviewer2 · 2020-02-13
**Mostly well executed, but borderline paper.**

**Rating:** 5
**Confidence:** 4

**Review:**



This paper presents a novel mechanism of providing text input in VR using a stylus on a tracked tablet. The paper describes their method of input and implementation, as well as the results of a user study comparing traditional keyboard input to their new approach, as well as an existing technique of raycasting to a keyboard.

Overall, I like the aim of this work. Text entry in VR is far from a solved problem. The paper was also very easy to read and the methods were well described.

I have a couple of concerns preventing me from recommending acceptance. The success of the technique and its benefit over prior approaches, the methods used for testing and analysis, and the presentation of the paper.

Regarding the technique itself, it seems like a straight-forward idea. However, it does require the user to hold a stylus and tablet, which can be bulky and may not be compatible with a lot of  VR tasks that require more unencumbered interactions. I would also be curious to see how this approach compares to a direct stylus-based text-input mechanism. I have not seen studies comparing the relative impact of being able to see the tablet directly, the impact of having to hold the tablet in 3D space vs. Resting on a surface, or even stylus vs. finger input on these traditional interfaces. Instead, the study presented compared the novel technique against a pretty common, commercial baseline and found little difference in terms of resulting effect on text entry speed.

Regarding the methods, while the results are presented well, but it wasn't clear if it was a multi-factor repeated measures ANOVA that was run. It seems like there were independent one-way RM-ANOVAs run on trial and condition. There are no interaction effects mentioned, and so it's not clear exactly how the analysis was conducted, and if all multiple-comparison effects were accounted for.

Lastly, the presentation of the paper was a bit troublesome in some places. In the introduction, the reader doesn't need a paragraph explaining the overall structure of the paper. Similarly, when presenting the results, figures should be referenced in parentheses, with the focus of the sentences emphasizing the synthesis of the results. E.g., 'there was no significant effect of text entry type on error (Figure 8)', rather than 'Figure 8 shows'. This is a minor issue, but something that will improve the readability of the paper.

---

### Official Review · AnonReviewer1 · 2020-02-14
**An empirical study comparing two text entry techniques in VR**

**Rating:** 4
**Confidence:** 5

**Review:**

The submission presents an empirical user study with 24 participants that compares the  participants' text entry performance (speed and error rate) in VR in two conditions: 1) using a Vive controller with raycasting, and 2) using a stylus on a touch-sensitive surface (tablet+stylus). The results of the study report that participants had higher text entry speeds and lower error rates using the Vive controller than tablet+stylus. The submission contributes to existing knowledge about people's text entry performance with two existing VR text entry techniques.

The main strength of the paper is a good user study design, which follows most of the best practices. The study had 24 participants, which is more than most existing text entry studies in the lab. The work tackles an interesting and timely topic.

However, there are a number of weaknesses that must  be addressed prior to publication: 1) the contribution is unclear, 2) study design and data analysis have potential issues, and 3) there is no clear future work or takeaway points.

The submission motivates a need for new VR text entry techniques that address existing challenges, but it does not describe how the two techniques or the empirical study address these challenges. The submission compares two text entry techniques that have already been studied. It seems like the intention was  to show that tablet+stylus has a potential to outperform the controller condition (the baseline). However, it does not. This could be en interesting negative finding; however, it is possible that this is due to design issues with the tablet+stylus  and the limitations that  the submission points out in the limitations section. The information in the limitations section is not a contribution either because such design considerations are already known to the community (e.g., providing visual feedback that would map the user's hand to a virtual hand). Having a more robust implementation of tablet+stylus could provide more evidence that this technique is an  alternative to text entry using raycasting.

Although the study design is mostly commendable, it seems like participants performed only a small number of trials in a single session. In this case, it is usually better to average the typing speeds over trials and to compare the mean typing speeds and  their variances  using a  parametric test (e.g., a t-test). However, ideally, the study would have multiple sessions over days to show the learning effect of each technique. Also, the statistical analysis could be improved. To analyze speeds over sessions (or trials  which I do not suggest), the repeated measures ANOVA  requires a post-hoc test to find  where the differences are following evidence of a main effect of trial on text entry speed.  It is a bit odd that error rates passed normalcy and sphericity tests because error rates (hopefully) tend to be skewed towards 0.  The submissions should report the tests.

The submission does not contain clear takeaway points that would suggest what the reader should learn from the work. This is reflected in a very vague conclusion and future work that only suggests minor improvement to the tablet+stylus technique based on the existing knowledge about typing in VR.

In summary,  this work has  the potential to uncover new knowledge about how people entry text in VR that could inform the design of future text entry techniques. However, there are a number of weaknesses that need to be addressed prior to publication.  Thus, I encourage the authors to continue their work and I look forward to their next iteration.

---

### Meta-Review · Area_Chair1 · 2020-02-14

**Recommendation:** Reject
**Confidence:** 5

**Metareview:**

All reviews found the work interesting and timely, study design to be mostly correct, and literature review commendable. However, all reviews ask about the contribution of the work given that neither of the text entry techniques are novel. Although, conducting empirical  studies that compare different text entry  techniques is valuable  (even if they are reproducibility studies), all reviews point to issues  with  the tablet_stylus technique  that might have disadvantaged it in various ways. Also, reviews point to potential issues with statistical analysis. Thus, all reviews recommend rejecting this submission at this time.

---

### Decision · Program_Chairs · 2020-02-18

Reject